# Ophthalmological Approach for the Diagnosis of Dry Eye Disease in Patients with Sjögren’s Syndrome

**DOI:** 10.3390/life12111899

**Published:** 2022-11-15

**Authors:** Robinson T. Barrientos, Fernando Godín, Carlos Rocha-De-Lossada, Matias Soifer, José-María Sánchez-González, Esteban Moreno-Toral, Ana-Luisa González, Mike Zein, Pablo Larco, Carolina Mercado, Maria-Adelaida Piedrahita

**Affiliations:** 1Department of Ophthalmology, Instituto de la Visión, Quevedo 660003, Ecuador; 2Department of Ophthalmology, Research and Ocular Health Group, Unbosque, University of El Bosque, Bogota 110111, Colombia; 3Department of Ophthalmology, Qvision, VITHAS Almería Hospital, 04120 Almeria, Spain; 4Department of Ophthalmology, Regional Universitary Hospital of Málaga, 18014 Granada, Spain; 5Department of Surgery, Ophthalmology Area, University of Seville, 41012 Seville, Spain; 6Department of Opthalmology, Vithas Malaga, 29016 Malaga, Spain; 7Department of Ophthalmology, National Eye Institute, National Institute of Health, Bethesda, MD 20892, USA; 8Department of Physics of Condensed Matter, Optics Area, University of Seville, 41012 Seville, Spain; 9Department of Pharmacy and Pharmaceutical Technology, University of Seville, 41012 Seville, Spain; 10Department of Ophthalmology, Research Department Clínica La Luz, Lima 15046, Peru; 11Department of Ophthalmology, Bascom Palmer Eye Institute, School of Medicine, University of Miami Miller, Miami, FL 33136, USA; 12Department of Ophthalmology, School of Medicine, University CES, Medellin 050021, Colombia

**Keywords:** dry eye, Sjögren’s syndrome, evaporative dry eye, water deficiency dry eye, questionnaires, Schirmer I test, Schirmer II test, invasive tear film rupture time, tear meniscus height

## Abstract

Dry eye has two basic subdivisions: aqueous deficient dry eye (ADDE), with SS a major cause; and evaporative dry eye (EDE), due to either intrinsic or extrinsic factors. SS is a chronic inflammatory disorder defined by dysfunction of the exocrine glands leading to dry eye and dry mouth. The objective of this article was to carry out a systematic and critical review of several scientific publications on dry eye disease, with the aim of providing general recommendations to distinguish dry eye and its different variants in patients with SS, during the period 1979 to 2020, using search engines for articles indexed in Scopus, Latindex, Scielo, Clinical Trials, Medline, Embase, and Cochrane, allowing the analysis of 132 articles published in indexed journals on the subject of dry eye disease and SS, evidencing its conceptualization, prevalence, risk factors, etiopathogenesis, clinical manifestations, diagnosis, and treatment.

## 1. Introduction

Sjögren’s syndrome (SS) is a chronic, systemic autoimmune disease that causes the dysfunction of the exocrine glands. It includes immune-mediated damage to the lacrimal and salivary glands, destroyed by infiltrating lymphocytes [1,2]. Dry eye disease (DED) is a multifactorial disease of the ocular surface characterized by a loss of tear film homeostasis and accompanied by ocular symptoms in which the instability and hyperosmolarity of the tear film, inflammation and ocular surface damage, and neurosensory abnormalities play an etiological role [3,4,5,6]. Symptoms can vary from itching or sandy to burning and stinging sensation. Diagnosis begins with determining the dry eye essential nature: aqueous deficient dry eye (ADDE), or evaporative dry eye (EDE) [1]. SS depends on the ADDE division, furthermore ADDE cases need to be investigated as potential SS associated dry eye. In Figure 1, the leading etiological causes of DED are represented [3,4,5,6].

Since the 2017 TFOS DEWS II report, additional risk factors for SS have been identified, associating it with age, female gender, poor health, use of contact lenses, smoking, use of oral steroids or antidepressants, poorly managed thyroid disease and a greater extent of medical comorbidities, occupational risk factors (prolonged screen time), and environment factors (air conditioning and heaters) [5,7]. Factors that lowered the risk include a sedentary lifestyle and the use of angiotensin-converting enzyme inhibitors [8,9].

The human eye is usually protected from evaporation and desiccation by tear film homeostasis, which regulates the secretion of tears and distribution in the ocular surface in response to the blinking reflex. DED is characterized by a low quantity or quality of tears, destabilizing this microenvironment. The principal mechanism in DED is evaporative loss of water that leads to tearing hyperosmolarity. These mechanisms are thought to drive inflammation of the ocular surface and cell apoptosis in both the epithelial cells of the cornea and conjunctiva and the goblet cells of the conjunctiva [10]. In the case of SS, the etiology is not entirely clear. The presence of salivary gland epithelial cells expressing primary histocompatibility complex class II molecules and the identification of specific markers such as HLA-DR15 and HLA-DR3 imply that there are environmental antigens that trigger an inflammatory response [11]. The following is the recommended sequence (Table 1) of the diagnostic workup for DED:

### 1.1. Anamnesis

The patient should be asked about symptoms suggestive of dry eye. These include eye irritation, burning, stinging, foreign body sensation, blurred vision, improved vision when blinking, photophobia, or pain. The patient should also be asked about possible risk factors such as a history of collagen diseases, refractive surgery, stem cell transplantation, hepatitis, vitamin A deficiency, antihistamines, selective serotonin reuptake inhibitors, tricyclic antidepressants, and beta-blockers [9,12]. If the patient already has a diagnosis of SS, it is crucial to learn if this diagnosis was confirmed through gland biopsy or laboratory markers, Table 2.

### 1.2. Questionnaires on Symptoms

Numerous questionnaires are available for the assessment of symptoms in patients with DED, the TFOS DEWS II report recommends OSDI and DEQ-5 questionnaires. These questionnaires are both completed by the patient [7,9].

### 1.3. Determination of the Functional Visual Acuity (FVA)

The FVA is the continuous VA of the patient when performing activities such as reading or driving. However, standard VA tests may be normal [13].

### 1.4. Measurement of the Invasive Tear Film Breakdown Time (BUT)

The measurement of the invasive tear film breakdown time (BUT) is defined as the time between a complete blink and the appearance of the first tear film rupture. The average value is 10 s or more [14,15,16].

### 1.5. Lacrimal Osmolarity

Tear osmolarity was performed using TearLab^®^ (TearLab Corporation, San Diego, CA, USA). The cutoff suggestion ranges were 300–320 mOsm/L in mild DED, and 320–340 mOsm/L in moderate DED [17,18,19,20]. This test must be taken before the instillation of any drops or dyes on the ocular surface so it is the first thing that should be completed, with a differential value between eyes of > 8 mOsm/L) as a reference of positivity to specify dry eye [21,22,23].

It should be noted that the frequency of meibomian gland dysfunction (MGD) is higher in patients with SS than in the average population, which contributes to the worsening clinical picture. The complete diagnostic analysis for SS is used based on the European–American criteria (ACR/EULAR) proposed in 2016 for the classification of primary SS (Table 3) [24,25,26]. The more recent criteria are considered more refined and emphasize objective measures, including biopsy samples, as opposed to older criteria that were more exhaustive [25]. The biopsy sample study improves SS diagnosis but needs an interdisciplinary consultant with a rheumatologist [27,28,29]. When SS is suspected in a patient with dry eye, knowledge of the extraocular signs and symptoms of SS can aid in making the diagnosis [30,31,32].

### 1.6. Measurement of Matrix Metalloproteinases (MMP-9)

In recent years the technology has allowed clinical ophthalmologists, and other healthcare staff to measure MMP-9 levels in the tear film with a simple device, the InflammaDry assay (Rapid Pathogen Screening, Inc., Sarasota, FL, USA) [33,34]. The test is a rapid (<10 min) and noninvasive test that measures MMP-9 levels above 40 ng/mL. The results depend on two lines, one blue and one pink. Only the blue line on the test windows indicates a level under 40 ng/mL and two lines indicate levels above this mark [35].

### 1.7. Measurement of the Ocular Surface Sodium Fluorescein Staining

Following Sjögren’s international collaborative clinical alliance (SICCA) registry ocular examination protocol, the cornea staining score was determined [36]. Fluorescein stains areas of the ocular surface with corneal epithelial defects; its application allows for a better appreciation of the size of the tear meniscus. This technique should be performed before positioning the patient for evaluation and without lighting to avoid the lacrimation reflex and the formation of mucous filaments [37].

### 1.8. Dye Staining (Lissamine Green and Rose Bengal) of the Ocular Surface

Damage to the ocular surface can be examined with specific stains where defects of the corneal epithelium (rose bengal) and cell damage of the cornea and conjunctiva (lissamine green) can be seen and both must be performed separately [38]. In Table 4 the characteristics of the different colorants are compared [39].

The classification system first proposed by van Bijsterveld, on which current diagnostic criteria are based, qualifies three areas in each eye: the nasal and temporal bulbar conjunctiva and the cornea. The intensity of rose bengal staining is rated on a scale of 0 (no staining) to 3 (confluent staining) for each area (Figure 2). The maximum staining value for each eye is nine. Staining values of three or more are considered abnormal [8,40,41].

### 1.9. Tear Flow Evaluation Using the Schirmer I–II Test

The Schirmer test measures the basal and reflex tear secretion from the primary and accessory lacrimal glands and the volume of the marginal tear film. It is performed by placing a strip of special millimeter paper between the outer half of the lower eyelid and the bulbar conjunctiva of each eye, and then the patient keeps his eyes closed for 5 min. Then the strips of paper are removed, and it is observed how wet the paper is. Various cut-off values have been proposed for the diagnosis of DED, between ≤5 mm and ≤10 mm [7,42]. The test can be performed under topical anesthesia or without anesthesia.

### 1.10. Perform an Examination of the Ocular Surface

It is advisable to thoroughly inspect the external surface and evaluate the ease of expression of the meibomian glands and permeability of the glandular orifices in search of MGD. Patients with mild SS can present with a normal eye examination and preserved tear function. Severe SS can present with cicatricial conjunctivitis leading to fibrosis and scarring [1,3,5,6].

### 1.11. Evaluation of the Meibomian Glands (Meibometry)

The meibomian glands should be assessed for the volume, quality, and ease of expression of secretions. To assess the quality of secretion, it is measured with meibometry where each gland in the central zone of the lower lid is gently pressed, and the secretion of each gland is scored as 0 (clear or normal), 1 (cloudy), 2 (granular), and 3 (pasty, similar to toothpaste) [43]. The meiboscore is the assessment of the expressibility of the glands, it is studied by pressing five glands of the lower or upper eyelid with a cotton swab [44]. The score for the number of squeezable glands is 0 (all glands), 1 (three or four glands), 2 (one or two glands), and 3 (no glands) [43,45,46].

## 2. Treatment of DED

TFOS DEWS II recommends individualized management of DED due to lack of aqueous and evaporative secretion, as well as the overall severity of the disease [3,5]. A dry eye can indicate the presence of SS, particularly when it is associated with inflammation, difficulty in medical management, or dry mouth. A patient with suspected SS should have an interdisciplinary assessment and follow up. The first step to diagnosing and managing this disease is referring to a rheumatologist for systemic treatment. The patient should also be referred to a dentist for the prevention and management of oral diseases [47]. Treatment for DED progresses gradually, beginning with education, diet modification, eyelid hygiene, lubricating eye drops, environmental factors modification, and nonpharmacological, and pharmacological management (Table 5) [48,49,50].

Diet modifications to treat DED [51,52] include supplementation with essential fatty acids (i.e., omega-3, omega-6/gamma-linolenic acid, or both) [25]. In addition, increasing fluid intake to ensure adequate general hydration [5] and avoiding alcohol intake have been recommended [53].

With regard to complementary medicine, there is some evidence from clinical trials supporting the use of traditional Chinese herbs and acupuncture [51,52,53]. Using over-the-counter hot compresses, artificial tears, or other eye lubricants can be used as first-line treatments [4]. Measures for eyelid hygiene include detergent-based cleaning products and microblepharon-exfoliation procedures to help remove residue from the eyelid margin. Hygienic saline containing 0.01% pure hypochlorous acid has been shown to reduce biofilm, and hot compresses for eyelid hyperthermia are commonly used to soften the meibum and facilitate its exit from the ducts. Alternative options include topical corticosteroids for a limited duration, topical cyclosporine 0.05%, tacrolimus 0.03%, and lifitegrast 5%. Antibiotics such as oral doxycycline can also be given for two to three months [51,54,55,56].

Devices that can be used to preserve or stimulate tears include silicone-based punctal occlusion in thermolabile polymer and hydrogel devices, therapeutic contact lenses, and intranasal tear stimulation (e.g., TrueTear^®^, Allergan, Pleasanton, CA, USA). MGD can be treated by meibomian gland expression and devices such as vectorized thermal pulsation therapy (i.e., LipiFlow^®^, Johnson & Johnson Vision, Jacksonville, FL, USA), intense pulsed light (IPL, i.e., Optima IPL M22, Lumenis, Salt Lake City, UT, USA), light-based heat and compression (i.e., iLux, Alcon, Fort Worth, TX, USA), and portable thermal energy therapy (i.e., TearCare, Sight Sciences, Menlo Park, CA, USA) [4,47,57,58,59]. The third care step includes oral secretagogues and autologous or allogeneic eye drops. In contrast, step four includes topical corticosteroids for longer durations, amniotic membrane grafts, surgical punctal occlusion, and more complex surgical approaches.

## 3. Discussion

There is a high tendency in recent times to diagnose dry eye disease worldwide due to the multifactorial nature of this entity [60]. However, the casuistry collected in much of the literature shows that Caucasian populations with a prevalence of 0.04%, individuals older than 65 years, and females tend to have a more significant association. This low-rate result was from using autoantibodies to classify patients and according to Hochberg, it was 0.6% in Greece [43]. Similar results were found in Slovenia (0.6%), Denmark (0.6–21.11%), and the United Kingdom (3–4%) using the European criteria, however in the latter using the American–European consensus, the prevalence ranged from 0.1 to 0.4% [14,58]. Using the Copenhagen criteria, the prevalence was 2.7% in Sweden and 0.7% in China [14,60,61]. In Latin America, in Brazil it was 0.17% [62], Argentina it was 0.17% [63], using the COPCORD methodology, and in Colombia it was 0.12%, with the American–European classification criteria employed in all of these [64].

There are few demographic and characterization studies of SS in Latin America, while, in other countries, important information on its management, diagnosis, and treatment is available. In a study carried out in Colombia in 2016, with 58,680 cases, they found a prevalence in people over 18 years of 0.12%. Eighty-two percent were women, with a 4.6:1 female:male ratio and there was a higher prevalence among the 65 to 69 age group. In Ecuador, Oviedo and Moya (2019), reported a prevalence of dry eye disease in a population that varied between 27 and 88% according to the OSDI, McMonnies, and DEQ5 questionnaire, estimating a range of 27–34.5% with a median age of 34 years [64,65,66]. These findings are accentuated by the lack of timely medical attention and failure to reach a timely diagnosis [50].

In SS, dry eye and mouth have been reported in up to 30% of people over 65 years of age, particularly in women in their perimenopausal and postmenopausal years [67]. Ocular signs include hyperemia, conjunctival keratinization, punctate or filamentous keratitis, and in some cases, involvement of the eyelids [68]. At the ocular level, the examinations focus on the objective evaluation of tear production, stability, osmolarity, and evaluation of the lid margin and the ocular surface [69]. In general, ocular treatment includes artificial tears as the first treatment alternative in order to increase the volume of the tear film and reduce friction, topical corticosteroids, immunomodulatory agents, immunosuppressants, autologous serum, and in experimental studies, new treatments with cells are proposed (mesenchymal stem cells or multipotent stem cells (MSC)) [68], in addition to systemic treatments to treat extraglandular manifestations.

The relationship between some signs and symptoms in patients with Sjögren’s syndrome represents an example of “heterogeneity” associated with DED [26,70,71]. It is commonly associated with two possible etiologies: a water deficiency and excessive evaporation.

Although several studies have not reported a strong association or correlation between symptoms and signs in patients with dry eye assessed from the use of questionnaires and clinical tests [66] such as Begley et al. [45], Schein et al. [19], Hay et al. [63], Nichols et al. [32], and Lin et al. [72], most of them have found a weak or moderate association between these parameters, including a high correlation between the clinical diagnosis and the patient’s symptoms, suggesting that the symptoms have a more significant influence on the diagnosis of dry eye than the results of clinical tests [53].

Many other studies have reported an association or correlation between signs and symptoms of irritation in patients with dry eye due to meibomian gland disease or aqueous lacrimal deficiency, using various clinical tests such as Schirmer, BUT, fluorescein staining, clearing test of fluorescein, and corneal sensation (Afonso et al. [31], 1999; Macri & Pflugfelder, Pflugfelder et al. [39]) [16,64,73,74]. In terms of tear osmolarity, although it shows a very close agreement between the eyes and in the same eyes over time in normal subjects, it shows increasing variability in subjects with dry eye (Sullivan et al. [75] 2014). This is believed to be due to the instability of the tear film in affected patients and can be used as a diagnostic hallmark of DED [32,63,76]. These results advocate the clinical utility of a consensus of signs, which better captures the entire disease and discourages dependence on symptoms alone [77]. This finding differs from that reported by Schein et al. [19], who found no association between the presence of more frequent symptoms and a lower Schirmer result, regardless of whether the analysis was based on mean scores with a cut-off value of five or a cut-off value of seven [77]. Their sensitivity and specificity in the detection of symptomatic subjects was low [72,78,79].

Questionnaires are of some value for the evaluation of the etiology of dry eye [78,79]. However, alone they are insufficient to confirm a diagnosis of SS since they have no cost if they have a high value of sensitivity and specificity [13,80]. Tear function tests such as tear meniscus height and BUT play a role in the differential diagnosis of SS, the most important aspect being the difference between ADDE and EDE accompanied by the performance of the Schirmer I–II tests, as discussed above. The tear function index has been reported to be helpful in the diagnosis of SS [81,82,83,84]. In addition, vital rose bengal or lissamine green staining of the interpalpebral fissure is a noninvasive way to help diagnose SS, as conjunctival staining may be seen earlier in the course of the disease [85,86].

Regarding future lines of research, the noninvasive diagnostic tools for SS diagnosis and DED examination should be implemented in all ophthalmologist and optometrist scientific communities because of the the reliability and repeatability of the measurements. This allows the measurement of noninvasive breakup time which is more unambiguous to interpret than the one measured with fluorescein (as the FBUT might be related with local thinning of the tear film rather than with actual breakup event) and has been shown as potentially having a stronger correlation with patient discomfort. Furthermore, noninvasive techniques can be readily used by a broader range of medical personnel, such as technicians and nurses, which allows for more rapid and broadly available diagnostics of DED [87,88,89,90].

Another interesting future research line is the evaluation of the ocular surface microbiota in the pathogenesis and management of different eye diseases. Recently a larger multicenter study proposed the concept of eye community state type (ECST) with the aim of stratifying the different profiles of bacterial communities that coexist in a healthy eye. It was observed that nine different ECST could be considered within the healthy bacterial population [91]. However, the central job of the ocular surface and oral microbiota in the pathogenesis of SS is not completely understood, although microbiota changes have been distinguished [92] in these patients. Bacterial mimicry has been proposed as one of the systems by which the microbiome may take part in illness acceptance. Furthermore, microbiota dysbiosis in SS suggest that lower diversity may lead to higher disease activity [92]. Other studies have found different outcomes with greater phylogenetic diversity [93]. Therefore, commensal microbes could play a fundamental part in the pathogenesis of SS [94]. Peptides obtained from oral, stomach, and skin commensal microscopic organisms might prompt an insusceptible reaction by initiating the Ro60-receptive immune system microorganisms [94]. Likewise, it seems that an alteration of commensal bacteria in the gut caused a worsening of dry eye in SS. An improvement in microbiome health could improve the condition [93,95]. Nonetheless, the exact role of the role of the microbiota both in the management and in the diagnosis of this pathology should continue to be studied in multicenter studies with a larger number of patients.

## 4. Conclusions

Patient grumblings and clinical discoveries that are reminiscent of dry eye, particularly with ADDE, should always be considered possible indications of SS and a they should be given a brief further examination. Given the accessibility of new serologic indicative tests and the possibly extreme results of deferring a determination, the examination should include requests about corresponding side effects of oral dryness and a serologic assessment. Dry eye related with SS is not restricted to ADDE; associative MGD and EDE are frequently seen.

The ocular manifestations of SS are often accompanied by oral or systemic manifestations and a certain humeral profile. A severe dry eye accompanied by these systemic manifestations clarifies the diagnosis and directs a change in treatment from systemic treatment to a sequential ocular treatment.

There is no curative treatment available for SS, so a comprehensive treatment of the patient is essential: education and information, vigilance, and proactive steps by ophthalmologists and optometrists, in conjunction with rheumatology specialists, play a fundamental role and facilitate the early recognition of SS, allowing the management and timely intervention of ocular and systemic manifestations.

## Figures and Tables

**Figure 1 life-12-01899-f001:**
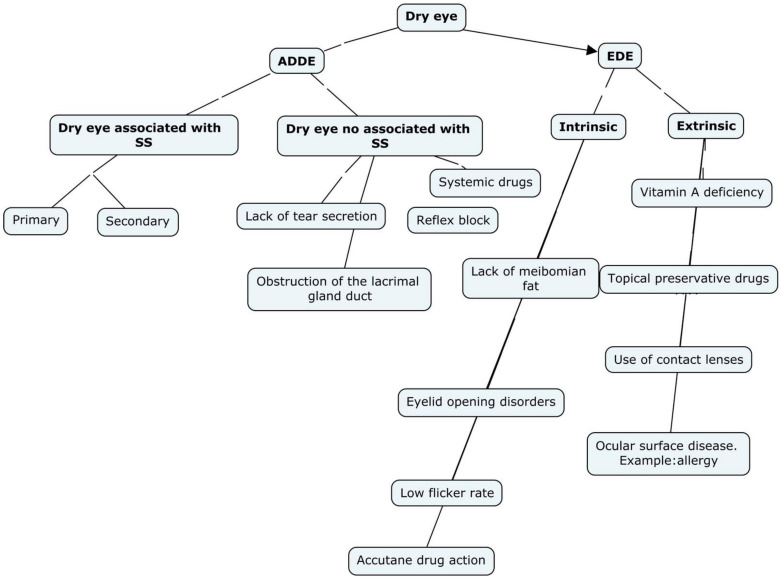
Main etiological causes of dry eye. Dry eye due to lack of aqueous secretion has two main groups: dry eye associated with SS and dry eye not associated with SS. ADDE: Aqueous deficient dry eye. EDE: Evaporative dry eye. SS: Sjögren Syndrome.

**Figure 2 life-12-01899-f002:**
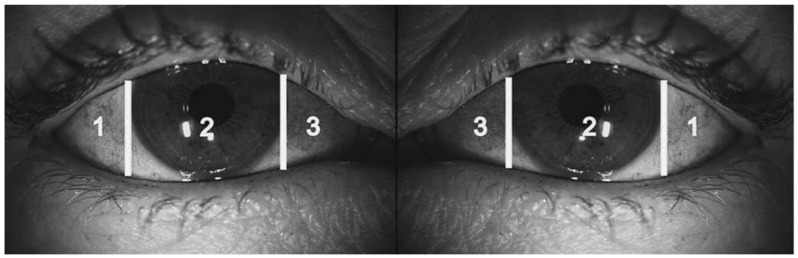
The exposed interpalpebral portions of the nasal and temporal conjunctiva and cornea are graded on a scale of 0 (no staining) to 3 (confluent staining). The maximum possible total score for each eye is nine. A score greater than three is considered abnormal.

**Table 1 life-12-01899-t001:** Sequence of diagnostic tests for DED.

History Use of symptom questionnaires. Determine functional VA and contrast sensitivity Measure the invasive tear film break-up time (BUT) Ocular surface staining with fluorescein sodium Ocular surface staining with lissamine green Evaluation of tear flow using the Schirmer I–II test Perform an ocular surface examination Meibometrics Measurement of tear osmolarity

BUT: invasive tear film breakup time; Schirmer I test; Schirmer test. (Adapted from Merayo L, J. Spanish guidelines for the treatment of eye disease. Consensus document. Spanish Society of Ocular Surface and Cornea. 2017; 4 (1): 33).

**Table 2 life-12-01899-t002:** Directed interrogation of ocular and medical history.

Ocular
Use of topical treatments: frequency, duration, effects, and whether or not they contain preservatives (artificial tears, glaucoma treatments, corticosteroids, antihistamines, vasoconstrictors and phytotherapy preparations, etc.). Use of contact lenses: frequency, and care. Allergic conjunctivitis. History of eye or eyelid surgery. History of ocular surface diseases. Facial paralysis.
**Medical history**
Smoking (including passive smoking). Facial and eyelid hygiene (products, techniques, and frequency). Dry mouth sensation. Fatigue. Joint and muscle pain. Use of systemic drugs (diuretics, antihistamines, hormonal treatments, antidepressants, antineoplastics and any anticholinergic drug, etc.). History of systemic inflammatory diseases (Sjögren’s syndrome, rheumatoid arthritis and systemic lupus erythematosus, etc.). Menopause. Trauma (mechanical, thermal, and chemical). Atopy. Head and neck surgeries or transplants. Chronic viral infections. Neurological disorders (Parkinson’s disease and trigeminal neuralgia, among others).

(Adapted from Merayo L, J. Spanish guidelines for the treatment of eye disease. Consensus document. Spanish Society of Ocular Surface and Cornea. 2017; 4 (1): 34).

**Table 3 life-12-01899-t003:** European–American criteria (ACR/EULAR) proposed in 2016 for the classification of primary SS.

Item	Score
Focal lymphocytic sialadenitis in minor salivary gland with ≥ 1 lymphocytic focus/4 mm^2^ of glandular tissue	3
Anti-SSA/Ro positive	3
Ocular staining score ≥ 5 (or ≥ 4 according to the Bjsterveld scale) in at least one eye	1
Schirmer test ≤ 5 mm/5 min, in at least one eye	1
Unstimulated salivary flow ≤ 0.1 mL/minute	1
Diagnosis > 4 points	

(Adapted from Shiboski CH, et al. Ann Rheum Dis 2017; 76: 9–16. doi:10.1136/annrheumdis-2016-210571).

**Table 4 life-12-01899-t004:** Comparison of the dyes fluorescein, rose bengal and lissamine green.

	Fluorescein	Bengal Rose	Lissamine Green
Healthy cell staining	No	Yes	No
Dead cell staining	No	Yes	Yes
Stain significance	Cell disruption and increase membrane permeability	Loss or insufficient protection of ocular surface mucin	Cell degeneration and death
Staining best seen with	Yellow barrier filter	Green barrier filter	Red barrier filter

(Adapted from Tseng SCG. Evaluation of the ocular surface in dry-eye conditions. Int Clin Ophthalmol 1994; 34: 57–69).

**Table 5 life-12-01899-t005:** Treatment schedule for DED.

Step	Treatment
Step 1	Education of the patient about: Management Possible dietary modifications Treatment Forecast Modifications of the local environment. Identification and modifications or elimination of systemic problems and topical medications Administration of warm compresses Perform eyelid hygiene and hot compresses Administration of eye lubricants
Step 2	If the options in Step one are inadequate: Manage ○ Unpreserved eye lubricants ○ Tea tree oil treatment for Demodex (if present) ○ Night treatments ○ Humidity chamber device ○ Ointments ○ Prescription drugs: doxycycline and tetracycline ○ Topical antibiotics or antibiotics and steroid combination ○ Topical corticosteroid ○ Nonglucocorticoid topical immunomodulatory-type drugs (such as cyclosporine) ○ Topical LFA-1 antagonist medication (such as Lifitegrast) ○ Oral macrolide or tetracycline antibiotics Preserve tears with punctal occlusion or moisture, camera glasses Complete therapies in the office ○ Intense Pulsed Light (IPL) therapy. Five sessions, with an interval of 3–4 weeks. Maintenance every 6–12 months. ○ Meibomian gland expression
Step 3	If the options in steps one and two are inadequate: Administer oral secretagogues Autologous or allogeneic serum eye drops Use of therapeutic contact lenses Soft bandage lenses Rigid scleral lenses
Step 4	If the options in the previous steps are inadequate: Administer topical corticosteroids for longer duration. Perform surgical options Amniotic membrane graft Surgical punctal occlusion Tarsorrhaphy Salivary gland transplant

Abbreviations: DEWS II, Dry Eye Workshop II; LFA-1, lymphocyte function-associated antigen 1; IPL, intense pulsed light. (Adapted from Jones L, Downie LE, Korb D, et al. TFOS DEWS II management and therapy report. Ocul Surf. 2017; 15: 575–628. Copyright © 2017 Elsevier Inc. All rights reserved).

## Data Availability

Not applicable.

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
