# Peer review of "Ophthalmological Approach for the Diagnosis of Dry Eye Disease in Patients with Sjögren’s Syndrome"

_life, 2022, doi:10.3390/life12111899_

Round 1

Reviewer 1 Report

The authors performed in the present manuscript a systematic review to provide actualized clinical guidance on the Sjögren's Syndrome diagnosis. I want to congrats the author for the work, the Dry Eye diagnosis and classification topic is highly interesting and, from my point of view, the issue asses here was well committed by the authors. In my opinion, the manuscript has minor flaws that should be enhanced previous to publication. In general, there are grammatical/formal

- I will suggest that authors carefully revise all the times that “Sjögren's Syndrome” in the whole manuscript, from “Simple Summary” to the conclusion section; sometimes both “Sjögren's” and “Syndrome was written with capital letters (ex. Lines 28, 31, 290, etc.) and in other times were written in lowercase letters (ex. Lines 38, 41, 290, etc.). In addition, the acronym “SS” was introduced in line 48, therefore, I will recommend using it in the entire manuscript and not on spare occasions.

- I will suggest replacing "dry eye disease/dry eye" in the whole manuscript (ex. Line 85) with "DED" (the acronym was previously introduced and used in the text on line 50) to unify the terminology.

- The author should correct the terminology for the test “Tear film break up time” in the entire manuscript, sometimes it is used as “BUT” (ex. Line 113) and sometimes is used as “TBUT” (ex. Line 26).

- Please, there is a minor problem of lack of references on some affirmations provided in the introduction (ex. Lines 74 to 77, lines 190 to 192); authors may enhance this issue.

- Simple Summary vs. Abstract vs. Main text: those two initial sections have some incompatibilities in the terminology with the rest of the manuscript (ex. Ocular Surface Disease Index Dry Eye Questionnaire (line 25 vs. Line 103). These differences seem to be generated because different authors wrote different sections.

- Figure 1: I want to suggest modifying the illustration to add the same terminology as in the rest of the manuscript (ex. Change or enhance the concept “Due to the lack of watery secretion” with the terminology previously used of “ADDE” to make easy reading of the author); I am aware that here the authors are talking about “causes” and no “types”, that this kind of accelerations will make the manuscript easy to read (Annotation: if new acronyms are added to the figure, it should be defined in the figure legend).

- Table 1: visual acuity vs. VA (line 109). Please, revise

- Line 119: I will recommend deleting the “USA” from “(TearLab Corporation, San Diego, CA, USA)” or adding it to the rest of the product quoted in the text, cause the rest of the manuscript only states are stated in those descriptions (ex. Lines 239, 247, 250, etc.).

- Line 126: This is the first time that the acronym is used, but it was not defined; it is described in line 192; please, correct.

- Line 313: I do not understand why the format of the reference in the text shows a change from this point (from this point, author and date are also reported with the reference number) (ex. 319-320); it makes me feel that the discussion was written by a different author.

From the above perspective, I think this manuscript should be accepted carefully.

Author Response

Reviewer 1

RV_1_01: The authors performed in the present manuscript a systematic review to provide actualized clinical guidance on the Sjögren's Syndrome diagnosis. I want to congrats the author for the work, the Dry Eye diagnosis and classification topic is highly interesting and, from my point of view, the issue asses here was well committed by the authors. In my opinion, the manuscript has minor flaws that should be enhanced previous to publication. In general, there are grammatical/formal.

AU_1_01: The authors thanks the comment and we have include all suggestion on the manuscript.

RV_1_02: I will suggest that authors carefully revise all the times that “Sjögren's Syndrome” in the whole manuscript, from “Simple Summary” to the conclusion section; sometimes both “Sjögren's” and “Syndrome was written with capital letters (ex. Lines 28, 31, 290, etc.) and in other times were written in lowercase letters (ex. Lines 38, 41, 290, etc.). In addition, the acronym “SS” was introduced in line 48, therefore, I will recommend using it in the entire manuscript and not on spare occasions.

AU_1_02: Comments: the corrections were made by the authors.

RV_1_03: I will suggest replacing "dry eye disease/dry eye" in the whole manuscript (ex. Line 85) with "DED" (the acronym was previously introduced and used in the text on line 50) to unify the terminology.

AU_1_03: Comments: THe correction was taken into account.

RV_1_04: The author should correct the terminology for the test “Tear film break up time” in the entire manuscript, sometimes it is used as “BUT” (ex. Line 113) and sometimes is used as “TBUT” (ex. Line 26).

AU_1_04: Comment: We made the change

RV_1_05: Please, there is a minor problem of lack of references on some affirmations provided in the introduction (ex. Lines 74 to 77, lines 190 to 192); authors may enhance this issue.

AU_1_05: Comment: It was referenced in the paragraph of the introduction

RV_1_06: Simple Summary vs. Abstract vs. Main text: those two initial sections have some incompatibilities in the terminology with the rest of the manuscript (ex. Ocular Surface Disease Index Dry Eye Questionnaire (line 25 vs. Line 103). These differences seem to be generated because different authors wrote different sections.

AU_1_06: Comment: Thank you for your recommendation.

RV_1_07: Figure 1: I want to suggest modifying the illustration to add the same terminology as in the rest of the manuscript (ex. Change or enhance the concept “Due to the lack of watery secretion” with the terminology previously used of “ADDE” to make easy reading of the author); I am aware that here the authors are talking about “causes” and no “types”, that this kind of accelerations will make the manuscript easy to read (Annotation: if new acronyms are added to the figure, it should be defined in the figure legend).

AU_1_07: Comment: The figure was remade with the recommendations.

RV_1_08: Table 1: visual acuity vs. VA (line 109). Please, revise

AU_1_08: Comment: The change was made

RV_1_09: Line 119: I will recommend deleting the “USA” from “(TearLab Corporation, San Diego, CA, USA)” or adding it to the rest of the product quoted in the text, cause the rest of the manuscript only states are stated in those descriptions (ex. Lines 239, 247, 250, etc.).

AU_1_09: Comment: The change was made

RV_1_10: Line 126: This is the first time that the acronym is used, but it was not defined; it is described in line 192; please, correct.

AU_1_10: Comment: The acronym was explained properly

RV_1_11: Line 313: I do not understand why the format of the reference in the text shows a change from this point (from this point, author and date are also reported with the reference number) (ex. 319-320); it makes me feel that the discussion was written by a different author.

AU_1_11: Comment: thank you for all your recommendations, they were all made to match your expectations.

RV_1_12: From the above perspective, I think this manuscript should be accepted carefully.

AU_1_12: Thank you very much for all your comments and suggestions.

Reviewer 2 Report

This is a thorough review on the diagnosis of dry eye disease (DED) in Patients with Sjögren's Syndrome and the risk factors associated with it based on 132 articles published in journals indexed in variety of reputable databases. The review summarizes well the definition of dry eye with the important update introduced by TFOS in 2017. The need to combine various methodologies in order to properly diagnose the type and the clinical severity of dry eye is highlighted well in the review. The various approaches based on grading the patient discomfort via variety of DED questionnaires as well as on objective clinical parameters (vital stains, invasive tear film break-up time (TBUT), Schirmer I-II test, lacrimal meniscus height, and an examination of the general ocular surface) is thoroughly emphasized.

Overall the manuscript is well written and represents a valuable summary of the current state of the knowledge in the field. My recommendation is related with what appears to be an important omission by the authors. Throughout the manuscript the importance of invasive tear film break-up time (TBUT), is repeatedly outlined. However an important trend in the diagnostics of the tear film and of the outer ocular surface is the implementation of non-invasive techniques. This allows to measure non-invasive breakup time which is more unambiguous to interpret than the one measured with fluorescein (as the FBUT might be related with local thinning of the tear film rather than with actual breakup event) and has been shown as potentially having stronger correlation with the patient discomfort. Furthermore non-invasive techniques can be readily used by broader range of medical personnel like technicians and nurses which allows for more rapid and broadly available diagnostics of DED. Whole body of literature exists on the application of keratography and of specular microscopy for the determination of non-invasive breakup time and pattern of tear film instability in Sjögren's Syndrome which deserves to be mentioned as a separate paragraph in the manuscript.

Author Response

Reviewer 2

RV_2_01: This is a thorough review on the diagnosis of dry eye disease (DED) in Patients with Sjögren's Syndrome and the risk factors associated with it based on 132 articles published in journals indexed in variety of reputable databases. The review summarizes well the definition of dry eye with the important update introduced by TFOS in 2017. The need to combine various methodologies in order to properly diagnose the type and the clinical severity of dry eye is highlighted well in the review. The various approaches based on grading the patient discomfort via variety of DED questionnaires as well as on objective clinical parameters (vital stains, invasive tear film break-up time (TBUT), Schirmer I-II test, lacrimal meniscus height, and an examination of the general ocular surface) is thoroughly emphasized.

Overall the manuscript is well written and represents a valuable summary of the current state of the knowledge in the field. My recommendation is related with what appears to be an important omission by the authors. Throughout the manuscript the importance of invasive tear film break-up time (TBUT), is repeatedly outlined. However an important trend in the diagnostics of the tear film and of the outer ocular surface is the implementation of non-invasive techniques. This allows to measure non-invasive breakup time which is more unambiguous to interpret than the one measured with fluorescein (as the FBUT might be related with local thinning of the tear film rather than with actual breakup event) and has been shown as potentially having stronger correlation with the patient discomfort. Furthermore non-invasive techniques can be readily used by broader range of medical personnel like technicians and nurses which allows for more rapid and broadly available diagnostics of DED. Whole body of literature exists on the application of keratography and of specular microscopy for the determination of non-invasive breakup time and pattern of tear film instability in Sjögren's Syndrome which deserves to be mentioned as a separate paragraph in the manuscript.

AU_2_01: Thank you very much for your revision and suggestions. We really agree with your non-invasive technique comments. We have included in the manuscript this point.

Reviewer 3 Report

The ocular manifestations of Sjogren's syndrome are often accompanied by oral or systemic manifestations and a certain humeral profile.A severe dry eye accompanied by these systemic manifestations clarifies the diagnosis and orders the treatment from systemic treatment to a sequential ocular  treatment.

The article presents a review about the ophthalmological approach to the manifestations of Sjogren's syndrome.The presentation has the general allure of a course that ends relatively abruptly , without clear and nuanced conclusions , that could provide solid arguments for a sequential therapeutic approach in these cases. The title of the article, which begins with an ophthalmological approach, suggests an approach to the disease from an ocular perspective. This is an approach with limited value in the diagnosis and control the disease.The diagnosis of Sjogren"s syndrome and treatment of ocular manifestations require an interdisciplinary oculo-rheumatological-stomatological immunological approach.The article does not show the synthesis of some experiences, which should simplify both the diagnosis and the treatment of the ocular manifestations of Sjogre"s syndrome.

Author Response

Reviewer 3

RV_3_01: The ocular manifestations of Sjogren's syndrome are often accompanied by oral or systemic manifestations and a certain humeral profile. A severe dry eye accompanied by these systemic manifestations clarifies the diagnosis and orders the treatment from systemic treatment to a sequential ocular  treatment. 

The article presents a review about the ophthalmological approach to the manifestations of Sjogren's syndrome. The presentation has the general allure of a course that ends relatively abruptly , without clear and nuanced conclusions , that could provide solid arguments for a sequential therapeutic approach in these cases. The title of the article, which begins with an ophthalmological approach, suggests an approach to the disease from an ocular perspective. This is an approach with limited value in the diagnosis and control the disease. The diagnosis of Sjogren"s syndrome and treatment of ocular manifestations require an interdisciplinary oculo-rheumatological-stomatological immunological approach.The article does not show the synthesis of some experiences, which should simplify both the diagnosis and the treatment of the ocular manifestations of Sjogren's"s syndrome.

AU_3_01: Thank you very much for your revision, your recommendations upgraded the manuscripts content to a more professional level, now the conclusions seem to be more organised and professional.
